

# First specific detection and validation of tomato wilt caused by *Fusarium brachygibbosum* using a PCR assay

Siyi Deng[1,2,3,*], Quanke Liu[4,*], Wei Chang[1,2,3], Jun Liu[1,2,3] and Hua Wang[1,2,3]

[1] Institute of Plant Protection and Soil Fertilizer, Hubei Academy of Agricultural Sciences, Wuhan, Hubei, China
[2] Key Laboratory of Integrated Pest Management on Crops in Central China, Ministry of Agriculture, Wuhan, Hubei, China
[3] Hubei Key Laboratory of Crop Disease, Insect Pests and Weeds Control, Wuhan, Hubei, China
[4] General Plant Protection Station of Hubei Province, Wuhan, Hubei, China
* These authors contributed equally to this work.

Corresponding authors
Jun Liu, liuj@hbaas.com
Hua Wang, wanghua4@163.com

## ABSTRACT

Tomato wilt is a widespread soilborne disease of tomato that has caused significant yield losses in many tomato growing regions of the world. Previously, it was reported that tomato wilt can be caused by many pathogens, such as *Fusarium oxysporum*, *Ralstonia solanacearum*, *Ralstonia pseudosolanacearum*, *Fusarium acuminatum*, and *Plectosphaerella cucumerina*. In addition, we have already reported that *Fusarium brachygibbosum* caused symptomatic disease of tomato wilt for the first time in China. The symptoms of tomato wilt caused by these pathogens are similar, making it difficult to distinguish them in the field. However, *F. brachygibbosum* specific identification method has not been reported. Therefore, it is of great importance to develop a rapid and reliable diagnostic method for *Fusarium brachygibbosum* to establish a more effective plan to control the disease. In this study, we designed *F. brachygibbosum*-specific forward primers and reverse primers with a fragment size of 283bp located in the gene encoding carbamoyl phosphate synthase arginine-specific large chain by whole genome sequence comparison analysis of the genomes of eight *Fusarium* spp.. We then tested different dNTP, $Mg^{2+}$ concentrations, and annealing temperatures to determine the optimal parameters for the PCR system. We evaluated the specificity, sensitivity and stability of the PCR system based on the optimized reaction system and conditions. The PCR system can specifically identify the target pathogens from different fungal pathogens, and the lower detection limit of the target pathogens is at concentrations of 10 pg/uL. In addition, we can accurately identify *F. brachygibbosum* in tomato samples using the optimized PCR method. These results prove that the PCR method developed in this study can accurately identify and diagnose *F. brachygibbosum*.

## INTRODUCTION

Tomato is a vegetable crop widely cultivated worldwide and plays a prominent role in global agricultural production and trade. In 2021, the global tomato cultivated

area was more than five million hm$^2$ with a production of 189 million tons (https://www.fao.org/faostat/en/#data/QCL). However, with the gradual expansion of tomato cultivation area and cultivation index, the damage caused by soil-borne tomato diseases has increased year by year, which has caused great economic losses to the tomato industry (Liang et al., 2015). Soil-borne diseases are seriously harmful in the field, the causes are complex and are often caused by combined infection with multiple pathogens that are difficult to distinguish in the field (Back, Haydock & Jenkinson, 2002). Common soil-borne diseases of tomato include fusarium wilt, bacterial wilt, verticillium wilt, brown root rot, etc., and their causal agents are Fusarium oxysporum f. sp. lycopersici, Ralstonia solanacearum, Verticillium dahliae and Pyrenochaeta lycopersici, respectively (Aragona et al., 2014; Chavarro-Carrero et al., 2021; Kwak et al., 2018; Zhou et al., 2022). These soil-borne diseases have caused severe economic losses in tomato growing regions.

Tomato wilt is a widespread soil-borne disease of tomato that mainly affects roots and stems, infects vascular bundles, impedes the transport of nutrients, and leads to tomato death due to lack of water and nutrients (Dean et al., 2012). The occurrence of tomato wilt has become increasingly serious, resulting in significant yield losses in many tomato growing areas around the world, and has become a bottleneck that hinders the sustainable development of the tomato industry (Ignjatov et al., 2012; Li et al., 2017). Currently, many reports show that tomato wilt is the most important soil-borne disease affecting tomato plants, such as tomato bacterial wilt caused by Ralstonia solanacearum and fusarium wilt caused by F. oxysporum f. sp. lycopersici (Dong et al., 2023; Elanchezhiyan et al., 2018; Li et al., 2021; Kwak et al., 2018; Zhou et al., 2022). However, many studies have shown that there are many kinds of pathogens that cause symptoms of tomato wilt, such as F. oxysporum (Choi et al., 2013), Ralstonia solanacearum (Sikirou et al., 2009), Ralstonia pseudosolanacearum (Garcia-Estrada et al., 2023), F. solani (Ajilogba & Babalola, 2013), F. acuminatum (Chai et al., 2018), Plectosphaerella cucumerina (Xu et al., 2014). In addition, our previous study was the first to report that Fusarium brachygibbosum causes the symptomatic disease of tomato wilt in China, resulting in leaf and root wilt of tomato (Liu et al., 2023). In the studied fields with a total area of 11.2 ha including 12 surveyed fields, disease incidence ranged from 40% to 70% (diseased tomato plants in each field/All tomato plants in each field), resulting in significant yield losses of tomato (Liu et al., 2023). Tomato wilt caused by F. brachygibbosum was similar to that caused by other pathogens, and it was difficult to distinguish them in the field.

The plant pathogen Fusarium spp., which causes host plant wilt in the field and Fusarium dry rot during storage, has become a global threat to crop growth, transportation, and storage (Tiwari et al., 2021a; Tiwari et al., 2021b; Tiwari et al., 2023). However, the control methods for Fusarium spp. also vary due to differences in infection and pathogenesis (Buttar et al., 2023). Currently, there are few studies on F. brachygibbosum, and the mode of infection and pathogenic factors are still unclear, so different control methods may be needed to prevent the development of the disease. Before formulating disease control measures, we first need to identify the diseases. However, the specific identification method of this pathogen has not yet been reported. Therefore, it is very important to develop a

rapid and reliable diagnostic method to identify *F. brachygibbosum* in order to formulate a more effective disease management program.

Rapid diagnostic tests will facilitate pathogen identification and lead to more effective management practices, such as guiding the proper use of fungicides before severe diseases occur (*Shen et al., 2010*). In recent decades, many molecular biology-based pathogen detection technologies have been gradually developed and applied in production practices. Compared with traditional detection methods based on isolation, cultivation and morphological observation combined with analysis of biochemical characteristics, molecular detection methods are time-saving and efficient, and have higher sensitivity and specificity. Molecular detection methods based on polymerase chain reaction (PCR) have been successfully used to detect many pathogens, *e.g.*, *Fusarium oxysporum* f. sp. *lycopersici* (*Inami et al., 2010*), *Ralstonia solanacearum* (*Schonfeld, Heuer & Van, 2003*), *Fusarium solani* (*Muraosa et al., 2014*), *Verticillium dahliae* (*Gayoso et al., 2007*), and *Alternaria solani* (*Kumar et al., 2013*). Therefore, developing a method to detect *F. brachygibbosum* can efficiently and accurately monitor the occurrence of the disease at different growth stages of tomato, and thus prevent and control tomato wilt caused by *F. brachygibbosum* in a timely manner.

In this study, we designed specific primers for *F. brachygibbosum* based on whole genome sequence comparison and developed a simple and efficient molecular PCR detection method by optimal parameters of PCR system, including dNTP, $Mg^{2+}$ concentration and annealing temperature. We then evaluated the specificity, sensitivity, and stability of the PCR system. The application of this detection method for the detection and analysis of tomato samples in fields can provide accurate detection of *F. brachygibbosum* and provide a simple and feasible method for the accurate diagnosis of tomato diseases.

# MATERIALS & METHODS

## Fungi strains and DNA extraction

The strain of *F. brachygibbosum* was identified by our laboratory (*Liu et al., 2023*). For the specificity tests, a total of 23 fungal pathogens were collected from Northwest Agriculture and Forestry University, Shaanxi, China; Nanjing Agricultural University, Nanjing, China; Yulin Normal University, Yulin, China; and Hubei Academy of Agricultural Sciences, Wuhan, China. All strains were routinely cultured in potato dextrose agar (PDA) plates (200 g$L^{-1}$ of potato extracts, 1% glucose, and 2% agar) and incubated for 7–10 days under 25 °C culture conditions. Genomic DNA of all strains was extracted from PDA plates using Plant DNA Kit (TIANGEN, Beijing, China) according to the manufacturer's instructions. All DNA samples were examined by spectrophotometer to check their quality and concentration and stored at −20 °C until use.

## Specific PCR primers design

The genome sequences of *F. brachygibbosum* HN-1 (GenBank accession number MU249523.1), *F. equiseti* D25-1 (GenBank accession number QOHM01000001.1), *F. graminearum* (GenBank accession number HG970332.2), *F. oxysporum* (GenBank accession number NC_030986.1), *F. proliferatum* Fp_A8 (GenBank accession number

MRDB01000001.1), *F. pseudograminearum* Class2-1C (GenBank accession number CP064756.1), *F. solani* JS-169 (GenBank accession number NGZQ01000001.1), *F. verticillioides* 7600 (GenBank accession number CM000579.1) were downloaded from the National Center for Biotechnology Information (NCBI) database. We then performed multiple alignments of the conserved sequences of all genomes using Mauve software (version 2.3.1) to obtain homologous sequences. Then, we used BioEdit software (version 7.0. 9.0) to align homologous sequences, and selected low homology regions from homologous sequences to design primers. The nucleotide sequence of the designed specific primers of the target strain were checked using the Basic Local Alignment Search Tool (BLAST) of the NCBI database to verify the homology between primer and sequence of the pathogen. The primer sets were synthesized by Sangon Biotech (Shanghai, China).

## PCR conditions optimization

The relevant parameters of the PCR system were optimized, including the annealing temperature of the primers, the concentration of dNTP, and the concentration of $Mg^{2+}$. The PCR was performed in a 25 μL reaction volume containing 0.125 μL of TaKaRa Ex Taq polymerase (5 U/μL), 2.5 μL of 10 × Ex Taq Buffe ($Mg^{2+}$ free), 0.5–4μL (0.5–4 mM) of $MgCl_2$ (25 mM), 1-8μL (0.1−0.8 mM) of dNTP mixture(2.5 mM each), template DNA 1.0μL (1ng/μL), upstream and downstream primers Fb-F/Fb-R 1.0 μL each. Finally, the volume of the reaction mixtures was filled up to 25 μL with sterilized double-distilled water. According to the optimal conditions of dNTP mixture and $Mg^{2+}$ concentration, PCR amplification was performed as follows: 95 °C for 5 min, 32 cycles of denaturation at 95 °C for 30 s, annealing at 50−70 °C for 30 s, extension at 72 °C for 1 min, and final extension for 10 min at 72 °C. Annealing temperature were set 12 temperature gradients (50, 51.1, 52.7, 55, 57.5, 60, 62.2, 64.4, 66.6, 68.4, 69.6, 70 °C), thus determining the optimal annealing temperature reaction conditions. The PCR products were viewed under UV light after being separated by electrophoresis in 1.5% agarose gels and stained with ethidium bromide solution.

## Sensitivity of the PCR assay

To verify the sensitivity of the PCR system, *F. brachygibbosum* DNA was diluted in a 10-fold gradient from 10 ng/μL to 10 fg/μL with sterile double-distilled water. Then, 1 μL of DNA dilution concentration was used as PCR template to test the detection limit of target pathogen by PCR. Seven concentrations of DNA were performed PCR amplified, and the amplification products were separated using 1.5% agarose gel, and the amplified products were detected by ethidium bromide solution.

## Specificity of the PCR assay

To evaluate the specificity of the PCR primers, PCR amplification was performed with optimized PCR systems using *F. brachygibbosum* DNA and DNA from 22 species of fungal strains, among including 15 *Fusarium* species, as templates. Subsequently, the PCR products were viewed under UV light after being separated by electrophoresis in 1.5% agarose gels and stained with ethidium bromide solution.

## Detection of the target pathogen within tomato samples from the field and artificially inoculated

To evaluate the practicality of the PCR detection method for *F. brachygibbosum*, we collected 12 field tomato samples from tomato growing areas. After a small piece of tissue was excised from the rootstock area of 12 tomato samples using a sterilized scalpel, genomic DNA was extracted from field tomato samples using the Plant DNA Kit (TIANGEN, Beijing, China) according to the manufacturer's instructions. In addition, for the artificial inoculation test, tomatoes were inoculated with conidial suspensions of each fungus ($1 \times 10^7$ spores/mL) in the rootstock area of each tomato. We inoculated six healthy tomato plants with *F. brachygibbosum* strain, and four tomatoe plants with sterile water. Extract genomic DNA using the Plant DNA Kit (TIANGEN, Beijing, China) according to the manufacturer's instructions. The DNA extracted from the field and artificially inoculated tomato samples were used as a template for the PCR assay, and the DNA from *F. brachygibbosum* served as a positive control, and the sterilized double-distilled water served as a negative control. PCR amplification was performed by an optimized PCR system. The PCR products were viewed under UV light after being separated by electrophoresis in 1.5% agarose gels and stained with ethidium bromide solution.

## RESULTS

### *Fusarium brachygibbosum* specific primers were designed by whole genome sequence comparison

By whole genome sequence comparison of *F. brachygibbosum*, *F. equiseti*, *F. graminearum*, *F. oxysporum*, *F. proliferatum*, *F. pseudograminearum*, *F. solani*, and *F. verticillioides*, we screened a pair of specific primers the detection of *F. brachygibbosum*, which were located in the gene encoding carbamoyl-phosphate synthase arginine-specific large chain (Fig. 1). The NCBI database does not contain whole genome annotation information of *F. brachygibbosum* HN-1 (GenBank accession number MU249523.1), so the locus tag of the gene encoding the carbamoyl-phosphate synthase arginine-specific large chain of *F. brachygibbosum* HN-1 cannot be determined. However, we determined by whole genome sequence comparison that the primer pair Fb-F/Fb-R is located in the genome of *F. brachygibbosum* HN-1 at nucleotide positions 7,241,876 to 7,241,896 and 7,242,138 to 7,242,158, respectively. The primer pair Fb-F (5′-CAATTGCTGCCACTCGACCTG-3′) and Fb-R (5′-TATTGTGGTGAGGAGGAGTCG-3′) for *F. brachygibbosum* was designed and synthesized, and the amplicon size of *F. brachygibbosum* was 283bp.

### Standardization of concentration of dNTP, Mg$^{2+}$ and annealing temperature for PCR system

Establishing the optimal parameters for PCR systems, including dNTP, Mg$^{2+}$ concentration, and annealing temperature, is key to improving PCR amplification efficiency. The results showed that at an annealing temperature of 57.5 °C, the bands of PCR products were the clearest and the detection effect was the best, indicating that 57.5 °C was the best annealing temperature (Fig. 2A). In addition, the band of amplified product was the clearest when 2μL dNTPs (Final concentration: 0.2mM) (Fig. 2B) and 1.5
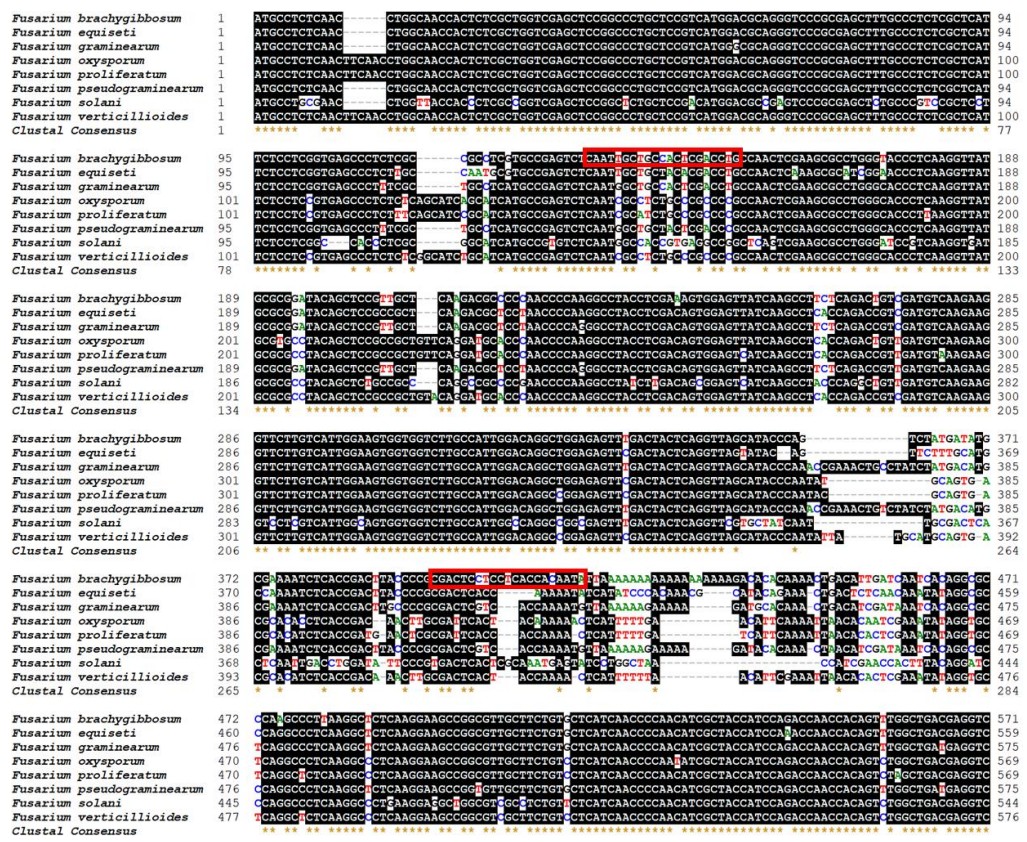

**Figure 1** **Location of primer sets for PCR detection of *F. brachygibbosum* strains.** Specific forward and reverse primers for *F. brachygibbosum* were developed using eight genomic regions of *Fusarium* spp. Homologous bases are shaded in black. Primer was marked with the read rectangle.

μL MgCl$_2$ (Final concentration: 1.5mM) (Fig. 2C) were added to the 25μl PCR system. Therefore, the optimal parameters for the 25 μL PCR assay system are: 0.125 μL TaKaRa Ex Taq polymerase (5 U/μL), 2.5 μL 10× Ex Taq buffer (Mg$^{2+}$ free), 1 μL forward primer, 1 μL for reverse primer, 1.5 μL MgCl$_2$ (25 mM), 2μL dNTP mixture (2.5 mM each), 1μL for DNA templates, and 15.875 μL ddH$_2$O. The amplification procedure of PCR was as follows: 5 min at 95 °C, followed by 32 cycles of 94 °C for 30 s, annealing at 57.5 °C for 30 s, and extension at 72 °C for 1 min. The final extension step was 10 min at 72 °C.

## Evaluation of PCR sensitivity

The extracted purified DNA of *F. brachygibbosum* was used to check the sensitivity of PCR using primers Fb-F and Fb-R. To determine the sensitivity of PCR detection, we performed a 10-fold dilution test with seven gradients of genomic DNA at concentrations ranging from 10 ng to 10 fg. The results showed that PCR with primers Fb-F/Fb-R yielded positive results from 10 ng to 10 pg DNA, but no positive signal from 1pg to 10 fg DNA as template (Fig. 3). Therefore, sensitivity analysis showed that the PCR molecular detection method established in this paper had a minimum detection amount of 10 pg for *F. brachygibbosum*.

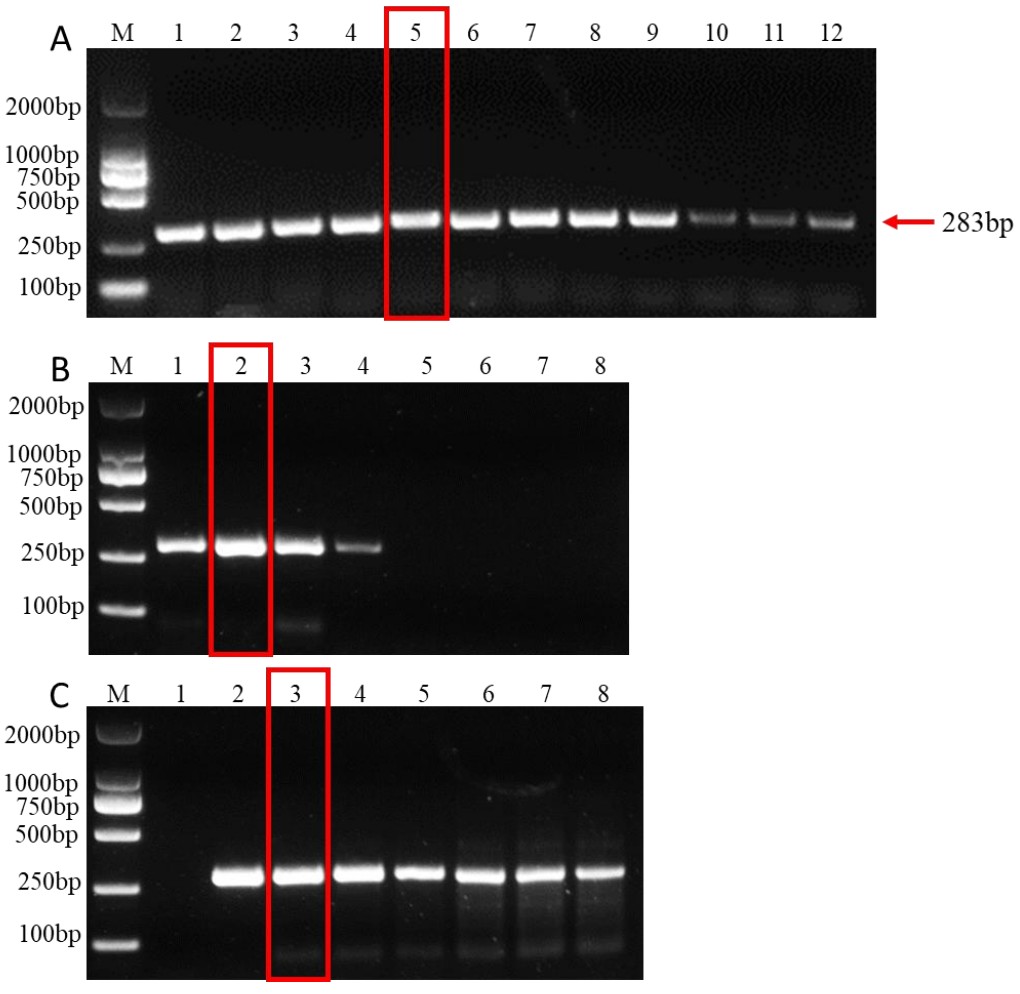

**Figure 2 PCR amplification at different PCR reagent composition and conditions.** (A) Gradients of annealing temperature. M: 2000 bp DNA ladder, Lane 1–12: 50, 51.1, 52.7, 55, 57.5, 60, 62.2, 64.4, 66.6, 68.4, 69.6 and 70 °C. (B) dNTP concentrations. M: 2000 bp DNA ladder, Lane 1–8: 0.1, 0.2, 0.3, 0.4, 0.5, 0.6, 0.7, 0.8 mM, respectively. (C) MgCl$_2$ concentrations. M: 2000 bp DNA ladder, lane 1–8: 0.5, 1, 1.5, 2, 2.5, 3, 3.5, 4 mM, respectively. The read rectangle indicates the optimal reaction system and conditions for PCR.

## Evaluation of PCR specificity

A total of 23 species of fungi were examined in PCR with *F. brachygibbosum*-specific primers Fb-Fand Fb-R (Table 1). The result of specificity detection showed that only the isolate of *F. brachygibbosum* yielded a product of 283 bp with primers Fb-Fand Fb-R, whereas 15 *Fusarium* spp. and other seven fungi no amplification product with primers Fb-Fand Fb-R (Fig. 4).

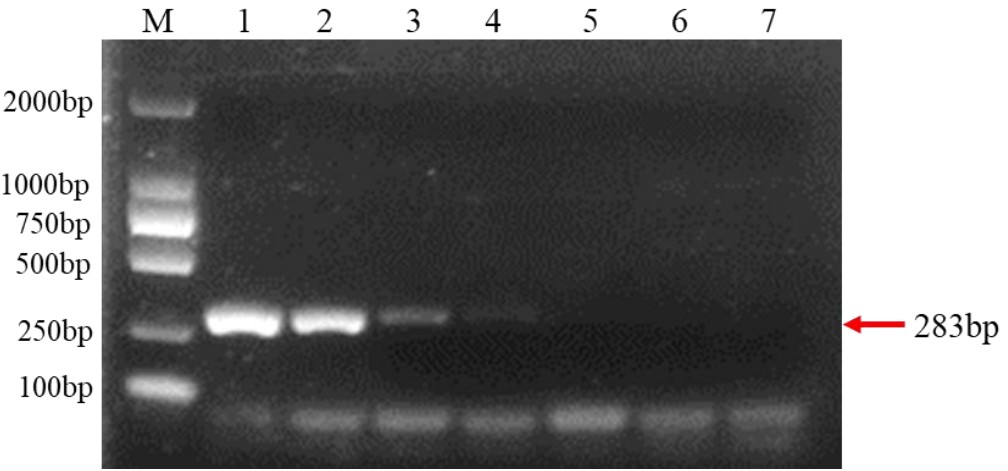

**Figure 3 Results of PCR sensitivity assay.** Sensitivity of molecular detection using the *F. brachygibbosum*-specific primers Fb-F/Fb-R designed in this study. M: 2000 bp DNA ladder; Lane 1–7: 10 ng/$\mu$l, 1 ng/$\mu$l, 100 pg/$\mu$l, 10 pg/$\mu$l, 1 pg/$\mu$l, 100 fg/$\mu$l, and 10 fg/$\mu$l pure genomic DNA.

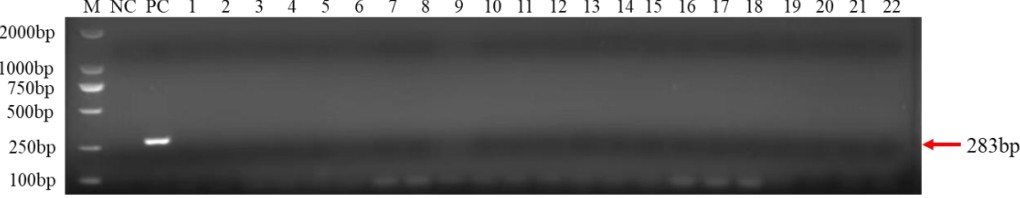

**Figure 4 Results of PCR specificity assay.** *F. brachygibbosum*-specific primers Fb-F/Fb-R only amplified the *F. brachygibbosum* strains DNA. M: DL2000 marker, NC: ddH$_2$O, PC: DNA of *F. brachygibbosum*, Line 1–22: DNA of *F. graminearum*, *F. verticillioides*, *F. solani*, *F. incarnatum*, *F. pseudograminearum*, *F. proliferatum*, *F. equiseti*, *F. oxysporum*, *F. oxysporum*, *F. oxysporum*, *F. oxysporum*, *F. oxysporum*, *F. oxysporum*, *F. asiaticum*, *F. fujikuroi*, *Verticillium dahliae*, *Alternaria alternata*, *Sclerotium rolfsii*, *Sclerotinia sclerotiorum*, *Rhizoctonia cerealis*, *Phomopsis amygdali*, *Botrytis cinerea*.

## The target pathogen within tomato samples from field and artificially inoculated were successfully detected by PCR

To test the practicality of this PCR assay, we detected 12 tomato samples collected from tomato producing areas where *F. brachygibbosum* pathogen was previously identified. The results showed that *F. brachygibbosum* was identified in eight of the 12 tomato samples from the field, among eight tomato samples (Fig. 5: Line 2, 3, 4, 5, 7, 9, 10, and 12) yielded a product of 283 bp, and four tomato samples no amplification product (Fig. 5: Line 1, 6, 8, and 11). In addition, the results of 10 tomato samples inoculated by artificial inoculation showed that the PCR results were consistent with those of inoculation, among six tomato samples (Fig. 5: Line 15, 17, 18, 20, 21, and 22) yielded a product of 283 bp, and four tomato samples no amplification product (Fig. 5: Line 13, 14, 16, 19).

**Table 1  List of fungi strains used in the specificity verification experiment of the multiplex PCR detection.**

| Strains[a] | Host species | Source[b] | Amplification result[c] |
|---|---|---|---|
| *Fusarium brachygibbosum*\* | Tomato | Isolate | + |
| *Fusarium graminearum* | Wheat | HBAAS | − |
| *Fusarium verticillioides* | Wheat | HBAAS | − |
| *Fusarium solani* | Tomato | HBAAS | − |
| *Fusarium incarnatum* | Rhizoma atractylodis | HBAAS | − |
| *Fusarium pseudograminearum* | Wheat | NWAFU | − |
| *Fusarium proliferatum* | Wheat | HBAAS | − |
| *Fusarium equiseti* | Pepper | HBAAS | − |
| *Fusarium oxysporum* | Wheat | HBAAS | − |
| *Fusarium oxysporum* | Tomato | YLNU | − |
| *Fusarium oxysporum* | Pepper | HBAAS | − |
| *Fusarium oxysporum* | Watermelon | HBAAS | − |
| *Fusarium oxysporum* | Tobacco | HBAAS | − |
| *Fusarium oxysporum* | Cucumber | HBAAS | − |
| *Fusarium asiaticum* | Rhizoma atractylodis | HBAAS | − |
| *Fusarium fujikuroi* | Rice | HBAAS | − |
| *Verticillium dahliae* | Tomato | NWAFU | − |
| *Alternaria alternata* | Tomato | HBAAS | − |
| *Sclerotium rolfsii* | Pepper | HBAAS | − |
| *Sclerotinia sclerotiorum* | Cauliflower | HBAAS | − |
| *Rhizoctonia cerealis* | Wheat | JAAS | − |
| *Phomopsis amygdali* | Peach | NJAU | − |
| *Botrytis cinerea* | Strawberry | NJAU | − |

**Notes.**
[a] Asterisks (\*) indicate the target pathogens of the multiplex PCR.
[b] HBBAS, Hubei Academy of Agricultural Sciences; JAAS, Jiangsu Academy of Agricultural Sciences; NWAFU, Northwest Agriculture and Forestry University; NJAU, Nanjing Agricultural University; YLNU, Yulin Normal University; HBAAS, Hubei Academy of Agricultural Sciences
[c] Specificity test results of multiplex PCR detection are indicated as positive (+) and negative (−).

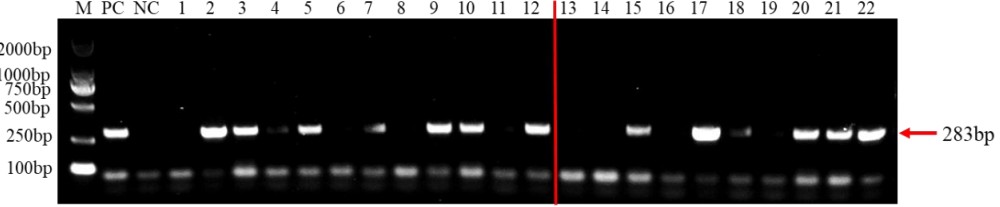

**Figure 5  PCR detection results of tomato samples from the field and artificially inoculated samples.**
M, 2000 bp DNA ladder; PC, *F. brachygibbosum*; NC, ddH$_2$O, Lane 1–12: tomato samples from the field. Lane 13–22: artificially inoculated tomato samples. Among, Lane 13,14,16,19: tomato samples were inoculated with sterile water.

## DISCUSSION

*F. brachygibbosum* has just been reported by our laboratory to also cause tomato wilt symptoms in China, resulting in significant tomato yield losses (*Liu et al., 2023*). Previous

studies have reported diseases caused by *F. brachygibbosum* in different hosts, *e.g.*, corn stalk rot (*Shan et al., 2017*), sunflower root wilt (*Xia et al., 2018*), onion basal rot (*Tirado-Ramírez et al., 2019*), and tobacco root rot (*Qiu et al., 2021*). Tomato wilt caused by *F. brachygibbosum* was similar to that caused by other pathogens, and it was difficult to distinguish them in the field. Currently, there is no report on specific detection of *F. brachygibbosum*, which leads to the inability to diagnose this pathogen quickly and accurately, so as to formulate prevention and control strategies. Therefore, it is necessary to develop rapid and efficient detection methods for *F. brachygibbosum* to prevent and control tomato wilt. In this study, we used whole genome sequence comparison to develop specific primers for *F. brachygibbosum* and constructed a PCR assay to accurately detect whether the pathogen causing tomato wilt is *F. brachygibbosum*.

PCR-based assays are widely used in fields such as ecology, environmental science, and agronomy to detect and monitor microorganisms in soil and plant (*Carnegie, Choiseul & Roberts, 2003*; *Cullen et al., 2001*; *Steffan & Atlas, 1991*). However, currently there is no PCR test available for the detection of *F. brachygibbosum*. *Fusarium* spp. has a wide range of classifications and high homology of genome sequences. With the rapid development of bioinformatics, whole genome sequence comparison can be used to find the difference regions between highly homologous pathogens and screen new pathogen detection targets, which provides a simpler and more efficient choice for establishing PCR system (*Hu et al., 2020*; *Kim et al., 2015*; *Park et al., 2017*). In this study, we compare eight genomes of *Fusarium* spp. to obtain homologous sequences. We used Bioedit software to compare homologous sequences of eight *Fusarium* spp. and selected different regions with low homology from the homologous sequences to design primers. We found that the regional sequence of *F. brachygibbosum* in Fig. 1 was specific and suitable for designing specific primers. This whole genome sequence comparison method is simple and easy to screen new targets for pathogen detection, which is a more efficient choice for establishing PCR detection system in the future. In addition, the region of specific primers designed in this study will also serve as a reference for adding TaqMan probe and designing qRT-PCR detection primers for quantitative detection of *F. brachygibbosum*. Therefore, in this study, we designed a primer pair (Fb-F and Fb-R) for the identification of *F. brachygibbosum* based on the whole genome alignment of eight common *Fusarium* genomes.

PCR reagent composition and PCR conditions are key factors affecting the amplification effect of PCR systems (*Markoulatos, Siafakas & Moncany, 2002*). PCR systems with a dNTP concentration of 0.2–0.4 mM are generally most favorable for the amplification reaction, and amplification is rapidly inhibited beyond this value, whereas a lower dNTP concentration (dNTP at 0.1 mM) allows PCR amplification but the amount of amplified product is significantly reduced (*Markoulatos et al., 1999*; *Markoulatos, Siafakas & Moncany, 2002*). In addition, optimization of the $Mg^{2+}$ concentration is critical because Taq DNA polymerase is a magnesium-dependent enzyme (*Markoulatos, Siafakas & Moncany, 2002*). Besides Taq DNA polymerase, both template DNA primers and dNTPs must bind $Mg^{2+}$. Too high a concentration of $Mg^{2+}$ stabilizes the DNA double strand and prevents complete denaturation of the DNA, reducing amplification yield, while too low a concentration of $Mg^{2+}$ reduces the amount of amplified product (*Markoulatos, Siafakas &*

*Moncany, 2002*). Therefore, we optimized the dNTP concentration, MgCl$_2$ concentration and annealing temperature in the system to improve the detection performance. In this study, the best detection results were obtained when 0.2 mM dNTPs and 1.5 mM MgCl$_2$ were added to a 25 $\mu$L PCR system at an annealing temperature of 57.5 °C. These results indicate that the detection efficiency of PCR requires the combination of multiple reaction systems and conditions.

Agricultural field soils are complex ecosystems with diverse microbial communities, soil and plant samples usually contain a variety of microorganisms (*Torsvik & Ovreas, 2002*). Therefore, it is very important to determine the specificity and sensitivity of *F. brachygibbosum* by PCR. This research showed that the PCR primers amplified only the DNA of *F. brachygibbosum* specific test strains with the expected amplicon size, indicating that the designed primer sets had high specificity for detection of the target pathogens. The same results were obtained in the detection of field and artificially inoculated tomato samples, indicating that the primer pairs had high specificity for the detection of *F. brachygibbosum*. In terms of sensitivity, the sensitivity of this method for detecting DNA concentration is 10 pg/uL, which meets the requirements for qualitative detection of pathogens in production.

## CONCLUSION

We designed primer pairs for *F. brachygibbosum* based on whole genome sequence comparison and developed a PCR method for *F. brachygibbosum* identification that has practical applications. The detection technology established in this study enabled efficient and rapid detection of *F. brachygibbosum* in diseased tomato tissue for the first time. This PCR method can provide reliable information for the detection of *F. brachygibbosum* in the field for early diagnosis and provide the basis for disease prediction and prognosis.

### Funding

This work was supported by the Natural Science Foundation of Hubei Province (2022CFB852), the China postdoctoral science foundation (2022M711097), the Key Technology Research and Demonstration Project of Hubei Agricultural Science and Technology Innovation Center (2020-620-000-002-07), and the Hubei Academy of Agricultural Sciences youth foundation (2023NKYJJ13). The funders had no role in study design, data collection and analysis, decision to publish, or preparation of the manuscript.

### Grant Disclosures

The following grant information was disclosed by the authors:
Natural Science Foundation of Hubei Province: 2022CFB852.
China postdoctoral science foundation: 2022M711097.
Key Technology Research and Demonstration Project of Hubei Agricultural Science and Technology Innovation Center: 2020-620-000-002-07.
Hubei Academy of Agricultural Sciences youth foundation: 2023NKYJJ13.

## Competing Interests

The authors declare there are no competing interests.

## Author Contributions

- Siyi Deng performed the experiments, analyzed the data, prepared figures and/or tables, authored or reviewed drafts of the article, and approved the final draft.
- Quanke Liu performed the experiments, analyzed the data, prepared figures and/or tables, and approved the final draft.
- Wei Chang performed the experiments, analyzed the data, prepared figures and/or tables, and approved the final draft.
- Jun Liu conceived and designed the experiments, performed the experiments, analyzed the data, prepared figures and/or tables, authored or reviewed drafts of the article, and approved the final draft.
- Hua Wang conceived and designed the experiments, performed the experiments, analyzed the data, prepared figures and/or tables, authored or reviewed drafts of the article, and approved the final draft.

## Data Availability

  The raw data are available in the Supplementary File.

## Supplemental Information

Supplemental information for this article can be found online at http://dx.doi.org/10.7717/peerj.16473#supplemental-information.

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
