# Peer review of "First specific detection and validation of tomato wilt caused by Fusarium brachygibbosum using a PCR assay"

_PeerJ, doi:10.7717/peerj.16473_

## Round 0.1 · original submission · Major Revisions

The authors are requested to revise the manuscript as per the reviewers' suggestions.

Reviewer 1 ·

Basic reporting

This manuscript entitled "First specific detection and validation of tomato wilt caused by Fusarium brachygibbosum using a PCR assay" aimed to develop a PCR assay specific to a tomato pathogen Fusarium brachygibbosum. Primers were designed based on the comparison of multiple closely-related Fusarium genome sequences and the PCR system was optimized. PCR assay specificity and sensitivity were both validated/tested. Overall, the manuscript is in good shape. Minor revisions in the description of the methods are suggested. The discussion section should be strengthened. Improvement in English writing is also needed.

Experimental design

The research question is straightforward and the experiment design is solid. The PCR assay's sensitivity and specificity were both addressed by appropriate experiments. Some revisions are suggested in the methods description (please see my line-to-line comments).

Validity of the findings

The PCR assay was well-validated for specificity using artificially inoculated and field samples.

Additional comments

Below are my line-to-line comments:

Line 45: Please clarify the citation FAOSTAT.

Lines 65-67: One question for these different pathogens with similar caused symptoms is that, do they need different treatments? This could also be one reason for the necessity of the new PCR assay specifically for the tested pathogen and could strengthen the manuscript's meaning.

Line 72: Please clarify this sentence. What does the 40-70% account for?

Line 82: Sentence not written correctly.

Line 123: How many low homology regions were obtained? And why is the region in Fig. 1 chosen? Providing a diagram showing the low homology regions in the overall genome alignment would be very helpful.

Lines 126-128: The sentence is not clear.

Line 157: The author included both artificial inoculation and field samples. But in this paragraph, it reads like 10 out of 12 were innoculated. Please revise the writing to clarify the two settings of the validation experiments.

Line 165: How was the tomato genomic RNA extracted?

Line 173: Please revise the term usage of "comparative genomics" throughout the study. It should be "whole genome sequence comparison". Comparative genomics refers to much broader analysis than just genome sequence alignment.

Line 194: Did the author test different cycle numbers?

Line 210: Do you mean collected 12 tomato samples? Are these 12 samples previously known of infection?

Line 220: I think the discussion section can be better extended. For example, why the specific region was chosen for primer design and how future studies can benefit from this method. Additionally, from Figure 1, it looks like it may be feasible to add a TaqMan probe and design a qPCR assay for the pathogen level quantification. Is this something that would benefit the field in the future?
The second paragraph in the discussion is repeated with the introduction and should be deleted/greatly shortened.

Reviewer 2 ·

Basic reporting

The article titled "First specific detection and validation of tomato wilt caused by Fusarium brachygibbosum using a PCR assay" is written in a clear tone with professional English. Experimental procedures have been reported in detail.

-Line 143: stained with nucleic acid dye. Authors could mention which dye was used to stain the bands.

-Line 150: Please reframe the sentence "amplification products are electrophoresis detected with 1.5% agarose gel" to "amplification products were separated using 1.5 % agarose gel, and the amplified products were detected by nucleic acid staining (Stained with EtBr or SYBR Safe?)

-Line 187: The authors wrote, "The results showed that at an annealing temperature of 53 °C, the bands of PCR products were the clearest and the detection effect was the best, indicating that 57.5 °C was the best annealing temperature". Could the authors clarify the best temperature, 53 °C or 57.5 °C?

-Line 185: Extension time was 1 min; however, the target amplicon size was ~250 bp. Could the author explain why a longer extension time was used for a shorter amplicon of 250 bp?

-Line 189: Instead of mentioning the uL of the dNTPs and MgCl2, please note the exact concentration either in mM or uM.

Experimental design

Methods in this manuscript are detailed well, which is sufficient to reproduce.

Validity of the findings

Significant findings have been described well. The results presented in this manuscript used a PCR-based method to precisely detect the Fusarium brachygibbosum DNA. The primer pair used in this study was able to distinguish the Fusarium brachygibbosum from the other Fusarium sub-species. Thus, this PCR-based assay could accurately detect Fusarium brachygibbosum from different sub-species. The results provided support the conclusion.

Additional comments

Authors could consider improving the readability of the manuscript by avoiding long sentences.

·

Basic reporting

This study addresses a critical issue in tomato farming by attempting to develop a specific and sensitive PCR-based diagnostic method for Fusarium brachygibbosum, a pathogen contributing to tomato wilt. The ability to accurately and quickly identify this pathogen is essential for effective disease management and control. The introduction could benefit from a more structured approach. Begin by briefly explaining tomato wilt as a problem, its economic significance, and the need for a precise identification method. Then, introduce Fusarium brachygibbosum and its significance in causing tomato wilt, followed by the gap in specific identification methods. Please add recent references related to Fusarium spp. in introduction part. For examples:
1. RK Tiwari, BM Bashyal, V. Shanmugam, MK Lal, Ravinder Kumar, S Sharma, Vinod, K Gaikwad, B Singh, R Aggarwal (2021). Impact of Fusarium dry rot on physicochemical attributes of potato tubers during postharvest storage, Postharvest Biology and Technology, 181, 111638. https://doi.org/10.1016/j.postharvbio.2021.111638
2. Buttar, H.S., Singh, A., Sirari, A., Anupam, Kaur, K., Kumar, A., Lal, M.K., Tiwari, R.K., Ravinder Kumar (2023). Investigating the Impact of Fungicides and Mungbean Genotypes on the Management of Pod Rot Disease Caused by Fusarium equiseti and Fusarium chlamydosporum Frontiers in Plant Science, 14: 1164245. https://doi.org/10.3389/fpls.2023.1164245
3. Tiwari, R.K.; Lal, M.K.; Kumar, Ravinder; Sharma, S.; Sagar, V.; Kumar, A.; Singh, B.; Aggarwal, R. (2023). Impact of Fusarium infection on potato quality, starch digestibility, in vitro glycemic response, and resistant starch content. Journal of Fungi, 9 (4): 466. https://doi.org/10.3390/jof9040466
4.Tiwari, R.K., Bashyal, B.M., Shanmugam, V., Lal M.K., Ravinder Kumar et al. (2021). First report of dry rot of potato caused by Fusarium proliferatum in India. Journal of Plant Diseases and Protection, 129: 173–179. https://doi.org/10.1007/s41348-021-00556-6

Experimental design

Please provide more detailed information about the comparative genomic analysis that led to selecting the gene encoding carbamoyl phosphate synthase arginine-specific large chain and the specific fragment chosen for primer design.

Clarify how the dNTP, Mg2+ concentrations, and annealing temperatures were determined. Were these based on a systematic optimization process, and what were the ranges tested?

Detail the steps and conditions for PCR amplification to ensure reproducibility of the results.

Validity of the findings

Include a graphical representation or gel images to demonstrate the PCR amplification results for specificity, sensitivity, and stability. Visualization would enhance the understanding of the results.

Provide statistical analysis, if applicable, to support claims of specificity, sensitivity, and stability. This would add credibility to the findings.

Additional comments

Discuss the study's limitations, potential sources of error, and how these could affect the accuracy of the diagnostic method. Transparency about the study's limitations is important for understanding the method's applicability.

Compare the developed PCR method with other existing methods, if any, for identifying Fusarium brachygibbosum. Highlight the advantages and disadvantages of this new method over others.

Address the potential challenges and practical applicability of implementing this PCR method in a real-world agricultural setting, considering factors like cost, ease of use, and equipment required.
Ensure the manuscript is well-proofread to eliminate grammatical errors and improve clarity of expression.

·

Basic reporting

The study highlights Fusarium brachygibbosum as a pathogen causing tomato wilt and leading to significant yield losses in China. Previous studies have also identified this pathogen in various hosts, causing diseases such as corn stalk rot, sunflower root wilt, onion basal rot, and tobacco root rot. The difficulty in distinguishing Fusarium brachygibbosum-caused tomato wilt from wilt caused by other pathogens necessitates the development of rapid and efficient detection methods to formulate prevention and control strategies. Polymerase Chain Reaction (PCR) assays are widely used in ecology, environmental science, and agronomy to detect and monitor microorganisms. However, a specific PCR test for detecting Fusarium brachygibbosum was lacking.

Experimental design

Comparative genomics was used to identify unique genomic regions of Fusarium brachygibbosum, aiding in the design of specific primers (Fb-F and Fb-R) for PCR-based detection. The study optimized the PCR conditions, including dNTP and MgCl2 concentrations, as well as annealing temperature, to enhance detection efficiency. The designed primers demonstrated high specificity and sensitivity for detecting Fusarium brachygibbosum in test strains and various tomato samples, indicating the potential of this PCR assay for specific pathogen detection.

Validity of the findings

The research aims to contribute to the development of effective methods for early detection and management of Fusarium brachygibbosum-induced tomato wilt, ultimately reducing crop losses.

---

## Round 0.2 · accepted · Accept

The authors have revised the manuscript as per the suggested lines and now it can be accepted.

Reviewer 1 ·

Basic reporting

This manuscript is now clear and in good shape for publication.

Experimental design

The experiment is well-designed and clear stated.

Validity of the findings

The PCR assay was well-validated for specificity using artificially inoculated and field samples.

Additional comments

The author has carefully addressed all my concerns.

Reviewer 2 ·

Basic reporting

The manuscript "First specific detection and validation of tomato wilt caused by Fusarium brachygibbosum using a PCR assay" has got a significant improvements. The authors incorporated the suggested modifications in the introduction, methods, and results section. Authors also added recent and relevant references.

Experimental design

Authors also improved the methodology section that will help to reproduce the experiments.

Validity of the findings

Conclusions were well defined with respect to the research question.

Additional comments

no comment

·

Basic reporting

The manuscript is improved and now it can be accepted in its current form.

Experimental design

NA

Validity of the findings

NA

Additional comments

NA

·

Basic reporting

The manuscript is quite improved and now it can be accepted.

Experimental design

NA

Validity of the findings

NA

Additional comments

NA